# Synthesis and direct assay of large macrocycle diversities by combinatorial late-stage modification at picomole scale

Sevan Habeshian [1], Manuel Leonardo Merz [1], Gontran Sangouard[1], Ganesh Kumar Mothukuri [1], Mischa Schüttel [1], Zsolt Bognár [1], Cristina Díaz-Perlas [1], Jonathan Vesin [2], Julien Bortoli Chapalay [2], Gerardo Turcatti [2], Laura Cendron [3], Alessandro Angelini [4,5] & Christian Heinis [1✉]

Macrocycles have excellent potential as therapeutics due to their ability to bind challenging targets. However, generating macrocycles against new targets is hindered by a lack of large macrocycle libraries for high-throughput screening. To overcome this, we herein established a combinatorial approach by tethering a myriad of chemical fragments to peripheral groups of structurally diverse macrocyclic scaffolds in a combinatorial fashion, all at a picomole scale in nanoliter volumes using acoustic droplet ejection technology. In a proof-of-concept, we generate a target-tailored library of 19,968 macrocycles by conjugating 104 carboxylic-acid fragments to 192 macrocyclic scaffolds. The high reaction efficiency and small number of side products of the acylation reactions allowed direct assay without purification and thus a large throughput. In screens, we identify nanomolar inhibitors against thrombin ($K_i = 44 \pm 1$ nM) and the MDM2:p53 protein-protein interaction ($K_d$ MDM2 $= 43 \pm 18$ nM). The increased efficiency of macrocycle synthesis and screening and general applicability of this approach unlocks possibilities for generating leads against any protein target.

[1] Institute of Chemical Sciences and Engineering, Ecole Polytechnique Fédérale de Lausanne (EPFL), CH-1015 Lausanne, Switzerland. [2] Biomolecular Screening Facility, Ecole Polytechnique Fédérale de Lausanne (EPFL), Lausanne, Switzerland. [3] Department of Biology, University of Padova, 35131 Padova, Italy. [4] Department of Molecular Sciences and Nanosystems, Ca' Foscari University of Venice, Via Torino 155, Venezia Mestre, Venice 30172, Italy. [5] European Centre for Living Technologies (ECLT), Ca' Bottacin, Dorsoduro 3911, Calle Crosera, Venice 30124, Italy. ✉email: christian.heinis@epfl.ch

Omics technologies have identified a number of new potential drug targets in recent years, illuminating new opportunities to address unmet medical needs. However, it is difficult or impossible to generate small-molecule ligands for many protein targets, such as those with flat, featureless surfaces or important protein-protein interactions (PPIs). These types of targets have earned a label of "undruggable"[1], and prominent examples include MYC transcription factors, RAS family proteins, and β-catenin[2]. Some compounds such as monoclonal antibodies (mAbs) can bind to such targets, though they cannot modulate intracellular targets due to their membrane impermeability, and they are not orally available. Macrocycles are a molecular class with the potential to bridge the gap between small molecules and mAbs. These ring-shaped structures can be engineered to bind challenging targets, as well as to cross cell membranes or even attain oral availability if they are small (well below 1 kDa) with a low polar surface area (below 200 Å$^2$)[3,4]. A number of naturally derived and de novo developed macrocyclic compounds have already been implemented as drugs. Prominent examples include tacrolimus, rifampicin, erythromycin, lorlatinib, and glecaprevir[5], which demonstrate the enormous therapeutic potential of macrocycles.

Various published approaches have been used to generate macrocyclic ligands against targets of interest, including diversity oriented synthesis[6], one-bead-one-compound[7,8], biological display methods such as encoding by phage DNA, mRNA or plasmid DNA[9–11], and DNA-encoding of chemical libraries[12–14]. A common challenge with chemically synthesized combinatorial libraries are product mixtures from incomplete reactions (e.g., a difficult macrocyclization step), which can complicate hit identification. Biological display methods yield cleaner products, but are limited by the large size of the ligands (i.e., >1 kDa), which reduces membrane permeability. In principle, high-throughput screening in microwell plates is robust with maximal flexibility in assay format and readout. However, macrocycle libraries provided by commercial vendors such as Asinex (10,091 compounds), ChemBridge (11,000 compounds), or Polyphor[15] are rather small, limiting the chances of finding hits, in particular with challenging targets. Target-focused macrocycle libraries, that could afford higher hit rates, are limited and even smaller in size. Classical combinatorial synthesis methods, that were developed and successfully used for producing and screening peptide and small-molecule libraries[16–18], were applied only to a limited extent for generating macrocycle libraries. Methods for generating libraries comprising tens to hundreds of thousands of structurally and chemically diverse macrocyclic compounds, ideally tailored for protein targets by containing target-specific building blocks (e.g. fragments with weak affinity for the target) are highly desired.

Previously, we generated target-tailored macrocyclic libraries by combinatorially cyclizing hundreds of $m$ short linear peptides with $n$ diverse linkers, in order to screen $m \times n$ crude macrocyclic products in microwell plates[19]. Inspired by studies using mosquito liquid handling[20,21] or contactless acoustic droplet ejection (ADE) technology[22–25] for synthesizing and screening small molecule libraries, we recently applied ADE for miniaturizing chemical reactions and screening of macrocycles at smaller scale[26]. However, the reaction yields varied substantially due to the challenging nature of the macrocyclization reactions we applied, and the small number of commercially available linker reagents (<20) limited the diversities of these libraries[19,26,27]. While the approach allowed identification of nanomolar binders to targets with defined binding pockets such as trypsin-like serine proteases[19,27], we struggled to generate high-affinity binders to more challenging targets such as protein-protein interactions, the best ones obtained being PPI inhibitors of p53:MDM2 PPI with rather weak potency in the low micromolar range[26].

Herein, we aim at developing much larger and structurally more diverse target-tailored libraries of macrocyclic compounds to generate good ligands to challenging targets such as PPIs. To achieve this, we conceive a strategy in which $m$ macrocyclic scaffolds are diversified via lateral groups with $n$ fragments using an acylation reaction (Fig. 1a). For example, 192 scaffolds are combinatorially reacted with 104 fragments to generate a library of almost 20,000 macrocyclic compounds. Executing the reactions at a picomole scale and in nanoliter volumes allows for miniaturizing screens (e.g., in 384 or 1536 well plates), as well as limits the scale at which the $m$ macrocyclic scaffolds have to be purchased or produced. We perform the reactions at high concentrations to drive the reactions and to enable >100-fold dilution, thus allowing sufficient reduction of organic solvent concentration for direct screening of crude products in the synthesis plates (Fig. 1b). The approach offers a particularly easy access to chemically and structurally diverse macrocycle libraries that can be applied to any protein target.

## Results

**Acylation of macrocyclic scaffolds in nanoliter volumes**. To diversify macrocyclic scaffolds in a chemically efficient and combinatorial fashion (Fig. 1a), we modified peripheral amines with carboxylic–acid-based fragments (Fig. 1c). $N$-acylation reactions are fast and high yielding, as demonstrated in the synthesis of DNA-encoded chemical libraries[28] or the optimization of small molecule ligands[29,30]. We tested this acylation by reacting the model scaffold 1 (Fig. 1c), which has a primary amine as a peripheral group (lysine sidechain; amino group shown in blue), with eight structurally diverse carboxylic acids 1–8 (Fig. 1d). We initially performed the reactions on a "large" scale in 4 μL reaction volumes using 40 nmol of scaffold. To efficiently convert the scaffold into the desired products, we reacted it with a 4-fold molar excess of carboxylic acid (final conc. of 10 mM scaffold, 40 mM carboxylic acid) in the presence of an activating agent ($N,N,N',N'$-tetramethyl-$O$-(1H-benzotriazol-1-yl)-uronium-hexafluorophosphate, HBTU) and base ($N,N$-diisopropylethylamine, DIPEA). We found full conversion of the scaffold after 3 h of reaction with all but one of the eight acids via LC-MS analysis (top numbers in Fig. 1d and Supplementary Fig. 1). Here, an excess of carboxylic acid over scaffold was chosen as the smaller carboxylic acid was anticipated to contribute less towards target-binding than the scaffold moiety, meaning that the excess was therefore unlikely to shroud the activity of the macrocycles during screening. On the whole, we expected the excess carboxylic acid, $N$-hydroxybenzotriazole (HOBt), and tetramethylurea, both byproducts of the acylation reactions, to be generally compatible with biochemical assays.

To scale down the synthesis by 50-fold (to 800 pmol), we subsequently optimized the abovementioned reaction conditions using the same cyclic peptide scaffold in 80 nL as compared to 4 μL reaction volumes, and we transferred reagents by acoustic dispensing in droplet increments of 2.5 nL instead of transferring by a pipette. In our first attempt to transfer the reagents by acoustic waves, we applied the same 4 μL-scale reaction conditions to reactions in 80 nL volumes, but this yielded incomplete acylation of the model scaffold 1 (middle number in Fig. 1d; Supplementary Fig. 1). We hypothesized that the low yields were due to DIPEA, as the base is relatively volatile and may partially evaporate during the acoustic transfer in 2.5 nL droplets. We tested non-volatile bases, such as 1,4-diazabicyclo[2.2.2]octane (DABCO), the sodium salt of HEPES, and $N$-methylmorpholine (NMM), the latter of which was volatile but could be used at higher concentration due to increased

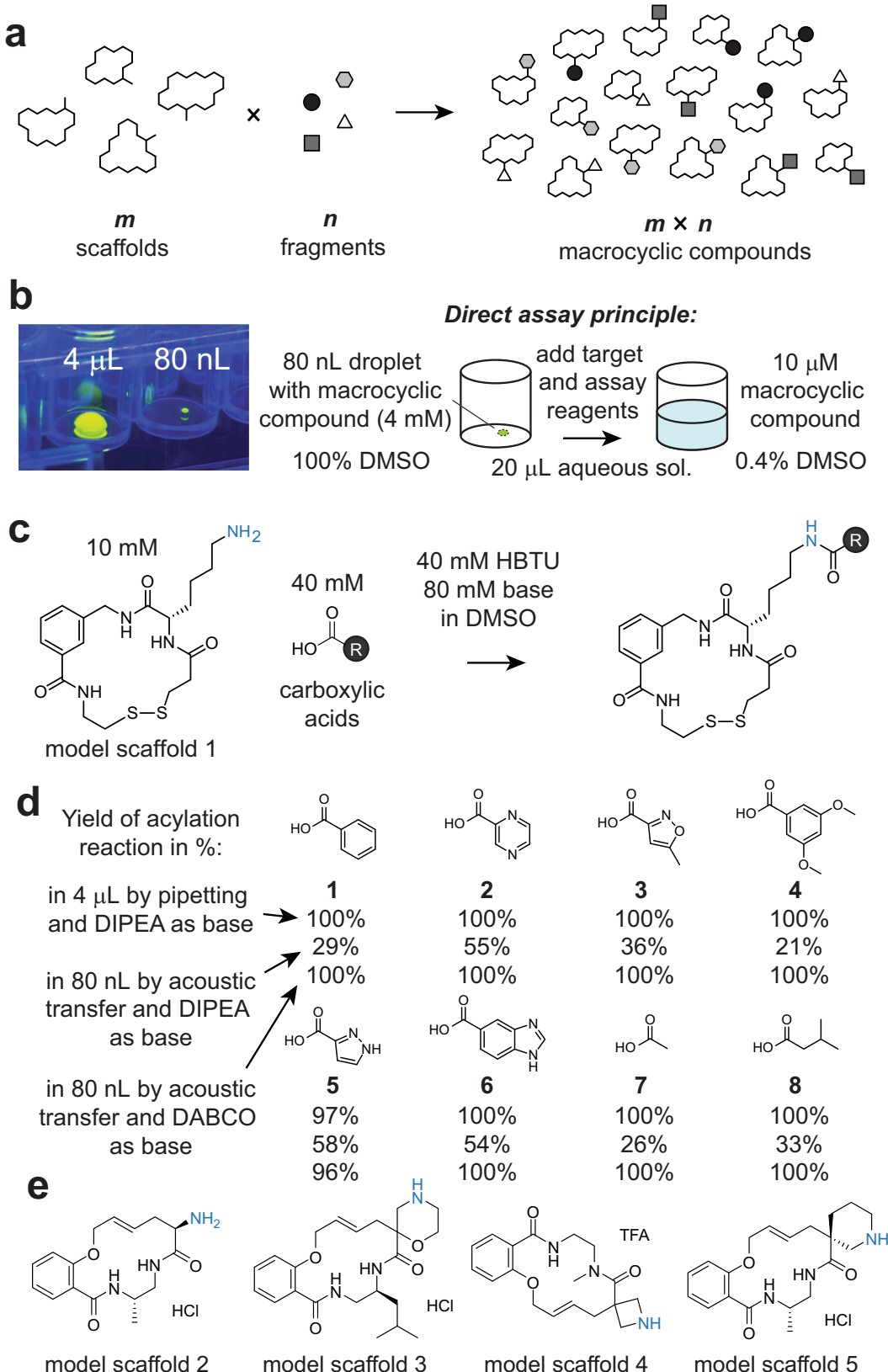

solubility in DMSO. We observed quantitative acylation of the macrocycles by test acids (**1**, **4**, **7**) using the new bases, despite the nanoliter droplet transfers (Supplementary Fig. 2). We chose DABCO for reactions with acids **1–8** and demonstrated quantitative modification of a peripheral primary amine at the nanomole scale (lowest number in Fig. 1d; Supplementary Fig. 3).

While the lysine ε-amine in model scaffold 1 is well accessible for acylation, primary or secondary amino groups closer to or within the macrocyclic backbones may be harder to modify. To assess the reactivity of such amines, we ordered four random macrocyclic scaffolds with primary or secondary amines from a commercial vendor (model scaffolds 2 to 5; Fig. 1e). We then

**Fig. 1 Diversification of macrocyclic scaffolds by combinatorially appending fragments to peripheral groups. a** General principle of the approach. **b** Image of an 80 nL droplet transferred by ADE, shown in a 96-well plate and next to a 4 µL droplet for scale. The droplets contain fluorescein for visualization by UV light. Addition of target and assay reagent to 80 nL macrocycle reactions dilutes the organic solvent to 0.4% which is compatible with bioassays. **c** Model macrocycle scaffold 1 containing a peripheral primary amine (blue) that is modified by acylation. **d** Reaction of model macrocycle 1 with indicated acids **1–8**, quantified by HPLC (absorbance and/or ion count). The first number indicates conversion at 4 µL volume via pipetting with DIPEA. The second and third numbers indicate conversion at 80 nL via acoustic liquid transfer with DIPEA and DABCO, respectively. **e** Randomly selected non-peptide scaffolds containing less accessible amino groups (in blue).

applied the same reaction conditions and carboxylic acids **1–8** at the picomole scale (80 nL volume, 800 pmol, DABCO as a base) (Fig. 1d). The model scaffolds 2, 3, and 5 were quantitatively acylated with all acids and scaffold 4 with most of them (acids **1**, **4–8**: 100%, acid **2**: 93%, acid **3**: 69%; Supplementary Fig. 4). These data indicated that the picomole-scale acylation is efficient at nanoliter volumes also for cyclic secondary amines.

**Scaffold and macrocycle library synthesis.** We next assessed whether the procedure for picomole-scale synthesis was suitable for generating a large macrocycle library. We synthesized 384 randomly selected cyclic peptide scaffolds that contained a peripheral amino group, using a recently developed approach for the production of small cyclic peptides in 96-well plates[31]. In brief, we synthesized short peptides on solid phase via a disulfide linker, removed the protecting groups, and released the peptide via a cyclative reaction, which yielded disulfide-cyclized peptides with 90% or greater purity (Fig. 2a). We synthesized scaffolds that contained three amino acids: one with a primary amine in the side chain (chosen from six amino acids); one α-amino acid with a random side chain (chosen from 15 amino acids); and one with a random backbone structure (chosen from six amino acids) (Fig. 2b and c; scaffold formats 1a-f). In a smaller sub-set of scaffolds, we introduced the primary amino groups through N-terminal cysteine residues (Supplementary Fig. 5; Scaffolds 1g-l). Of the 3,780 scaffolds that could theoretically be assembled based on the indicated formats and amino acid building blocks, we synthesized 384 randomly chosen ones, or 10%. We quantified cyclic peptides containing a Trp residue (a total of 45 scaffolds) by absorption measurements and found that we obtained most molecules in high yields and at a narrow concentration range (average yield = 1.6 µmol, corresponding to 32% based on resin loading; average conc. = 8.1 mM; Fig. 2d).

Next, we combinatorially reacted the 384 scaffolds with 12 carboxylic acids by acoustic dispensing and obtained 4,608 different macrocycles (Fig. 2e and Supplementary Fig. 6). As controls for the activity of the carboxylic acids alone, we performed an additional 384 reactions in which the scaffolds were incubated with all reagents but not carboxylic acids. To increase the speed of dispensing and to reduce scaffold consumption (160 pmol scale) relative to the test acylation reactions, we reduced the reagent transfer volumes slightly (20 nL of 8.1 mM scaffold, 20 nL of 80 mM pre-activated acid). We also added a greater excess of acid (10-fold; final concentrations of 4 mM scaffold and 40 mM carboxylic acid) to ensure quantitative modification of scaffolds with less accessible amines (Fig. 2f). Contactless liquid transfer for each 384-well plate took 5 min, and the synthesis of the 4,608-compound library required ~1 h. After incubation for 5 h, we quenched the reactions by the addition of 5 µL of 100 mM Tris-Cl buffer that reacted with excess activated acid overnight.

**Screening of 4608 macrocycles for thrombin inhibition.** As a proof-of-concept, we screened the macrocycle library for inhibitors against thrombin, which is the central blood coagulation

protease and an important therapeutic target. We conducted the screen for thrombin-inhibiting macrocycles in 384-well reaction plates. Briefly, each well contained a unique macrocycle (10 µM), 5 µL of thrombin (2 nM final conc.), and subsequently, 5 µL of a fluorogenic substrate to measure the residual thrombin activity and the extent of protease inhibition. We observed over 50% inhibition of thrombin in 0.2% (9 out of 4608) of the reactions (Fig. 3a). All inhibitory macrocycles contained the chlorothiophene carboxylic acid **14**. We confirmed the screen by repeating the macrocycle synthesis reactions for chlorothiophene carboxylic acid (384 scaffolds × acid **14**) and the subsequent thrombin screen, and found the same top hits, which demonstrated high reproducibility for both the picomole-scale diversification reactions and the activity screen (Fig. 3b).

Most scaffolds modified with acid **14** did not inhibit thrombin, and a control well with **14** only (no scaffold) did not inhibit either, suggesting that the acid fragment alone has a rather weak affinity for thrombin. Indeed, 5-chlorothiophene-2-carboxamide, mimicking the "side chains" of diamino acids acylated with **14**, inhibited thrombin only weakly ($K_i = 380\,\mu M$). The best three hits, M1, M2, and M3, were all structurally similar and based on scaffolds of the format cyclo(D3-B5-Xaa), wherein the amino acids Xaa were all α-amino acids with hydrophobic side chains (D-Val, L-Phe, L-Val; Fig. 3c). These data indicated that much of the binding affinity could be attributed to the specific structure of the cyclic peptide scaffolds and suggests that the excess of acid present in the screens is likely inert.

To assess whether the observed inhibitory activity derived from the macrocyclic compounds or reaction side products, we repeated the reaction of scaffold with carboxylic acid **14** for the best hits at a 250-fold larger scale (40 nmol). We then ran the reactions over a reversed-phase-HPLC column, separated the crude mixture into 20 fractions, which we lyophilized, and measured the thrombin inhibition activity in the lyophilized product from each fraction (Fig. 3d). We observed the highest activity in the fraction containing the desired macrocyclic product, which indicated that the hits were found in the screen based on activities of the macrocycles and not side-products. The fractionation of crude reaction products was applied for validating the approach, but it is not necessary to apply this procedure to hits found in future screens. The purified macrocycles M1, M2, and M3 inhibited thrombin with $K_i$s of $44 \pm 1\,nM$, $165 \pm 38\,nM$, and $125 \pm 8\,nM$, respectively (Fig. 3c, Supplementary Fig. 7). As the reducible disulfide bonds of the scaffolds screened herein are not desired for most therapeutic applications, we thus tested their replacement by non-reducible linkages. We synthesized derivatives of M1 with dithioacetal or thioether linkers (M4, M5; Supplementary Fig. 7) with $K_i$s of $83 \pm 8\,nM$ and $135 \pm 16\,nM$, respectively. Thus, we concluded that the disulfide bond could be replaced without much activity loss.

We performed crystallographic studies of M1 bound to human thrombin and X-ray analysis at 2.27 Å resolution (PDB 6Z48) (https://www.rcsb.org/structure/6Z48), which showed that M1 is an active-site inhibitor. The chlorothiophene group points into the S1 specificity pocket, and the macrocyclic ring fills the S2 (disulfide region) and S4 (valine) spaces (Fig. 3e and

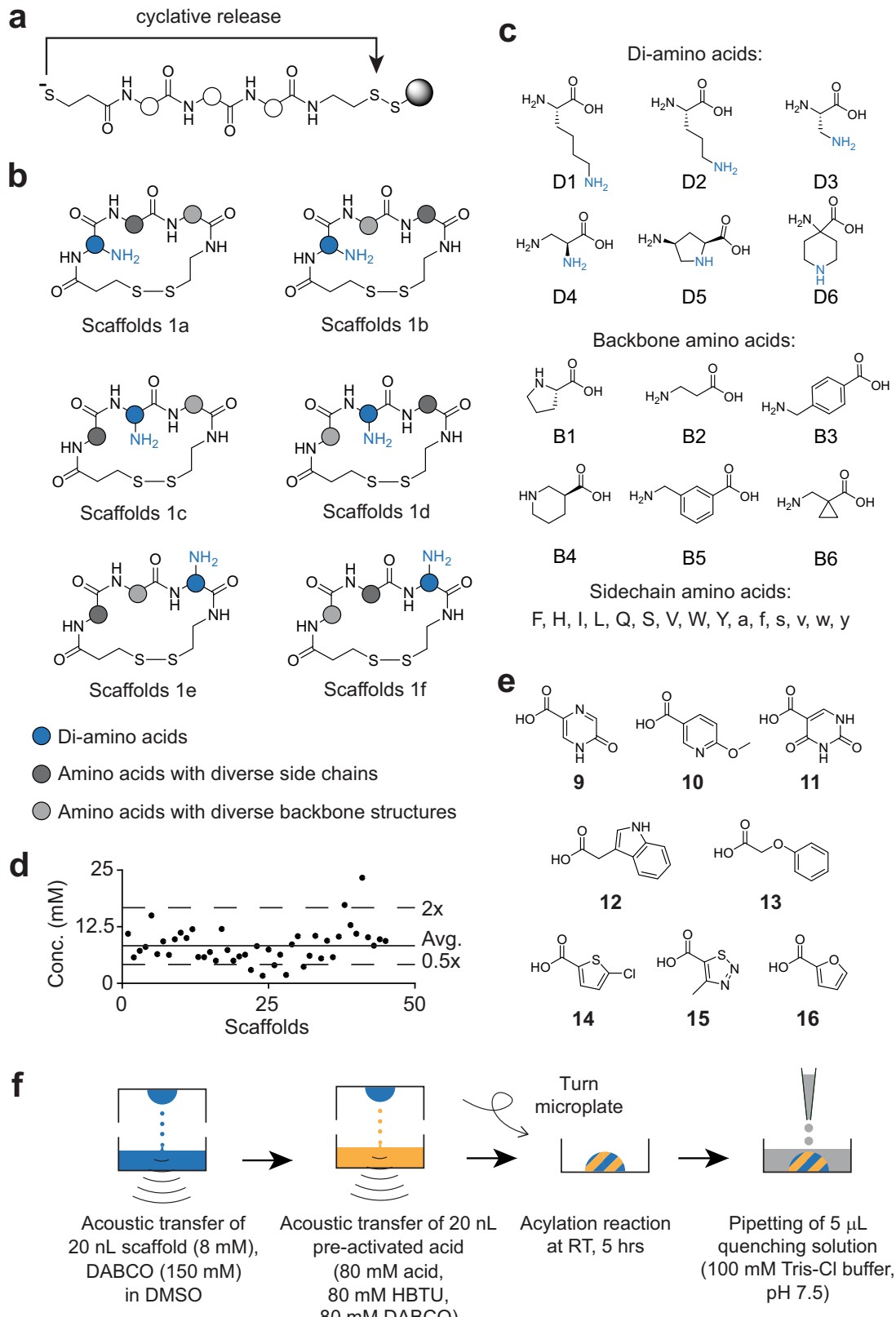

**Fig. 2 Acylation of amines in macrocyclic scaffolds and library generation. a** Cyclative disulfide release of side-chain-deprotected peptides. **b** Formats of cyclic peptide scaffolds. Amino groups are shown in blue. **c** Amino acids used for scaffold synthesis. Lower case letters indicate D-amino acids. **d** Yields of the 45 tryptophan-containing scaffolds as determined by absorption measurement. **e** Carboxylic acids **9–16** that were used with acids **1–3** and **8** to diversify the scaffolds shown in panel **b**. **f** Schematic of the protocol for macrocycle library synthesis by acoustic liquid transfer with indicated reaction conditions.

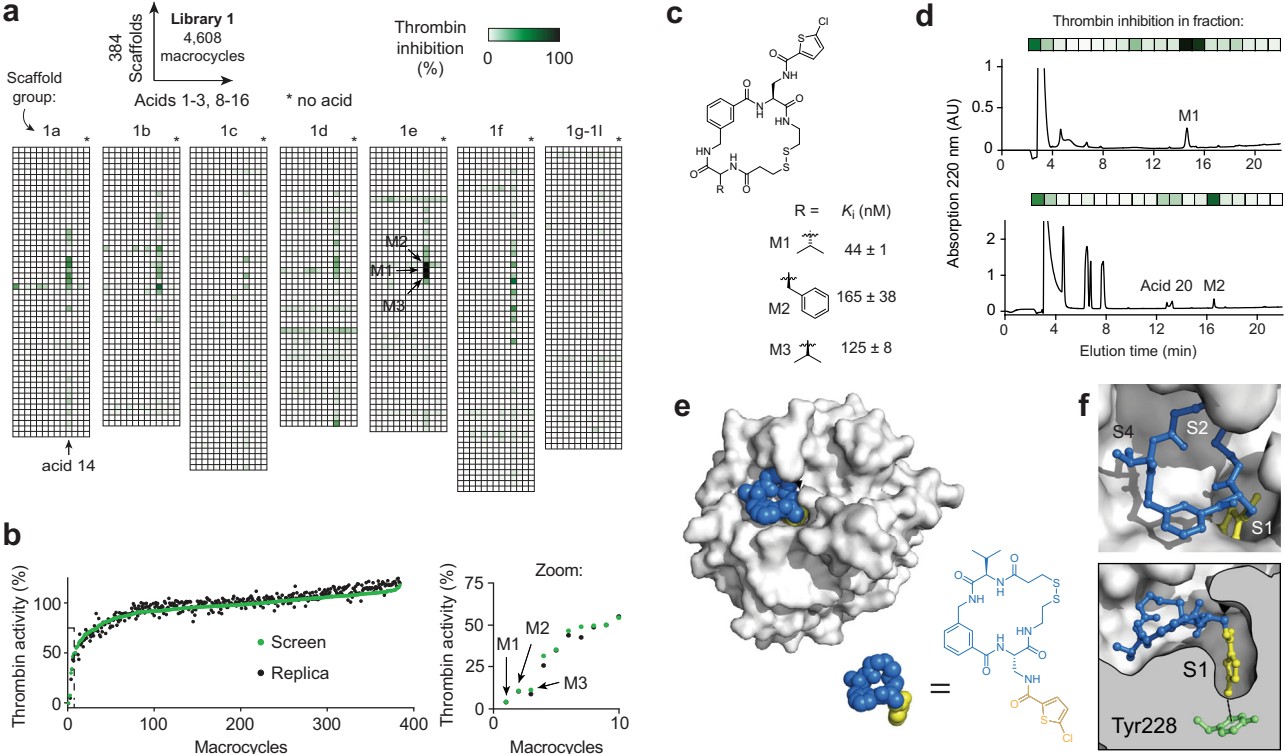

**Fig. 3 Screening of macrocyclic compound library against thrombin, hit identification, and structural analysis. a** Heat map with thrombin inhibition indicated for each macrocycle. The amino acid sequences of the scaffolds are provided in the Source Data file. **b** Replicate reaction (black) compared to original screen (green) of all macrocycles containing acid **14**. For each of the two independent experiments, the reactions were performed once and the thrombin inhibition was measured once. **c** Chemical structures and activities of the top three hits M1, M2, and M3. The mean values and SDs of three independent measurements are shown. **d** Chromatographic separation of the acylation reactions to generate M1 and M2 and analysis of the fractions for thrombin inhibition. **e** X-ray structure of M1 bound to human thrombin at 2.27 Å resolution (PDB 6Z48) (https://www.rcsb.org/structure/6Z48). The inhibitor is shown in the space-filling model with the scaffold in blue and the carboxylic acid in yellow. **f** Zoomed in structure of thrombin with the sub-sites indicated. The chlorothiophene group fills the S1 sub-site and forms an interaction with Tyr228.

Supplementary Fig. 8). The chlorothiophene group appears to form an aryl chloride-π interaction with a tyrosine residue at the bottom of the S1 pocket (Fig. 3f), similar to the previously reported binding of the Factor X inhibitor drug rivaroxaban that also contains a chlorothiophene group[32]. The structure of the M1:thrombin complex illustrated the contribution of both the macrocyclic scaffold and the carboxylic acid fragment to the overall binding and inhibition of thrombin. Thus, our crystallographic analysis further validated the approach of screening large macrocyclic libraries with diversified peripheral groups.

**Protein-protein interaction inhibitors and MDM2 screen**. We turned our efforts towards the search for macrocycles that can inhibit PPIs, which are often considered undruggable targets in disease. A prototypical PPI disease target is MDM2:p53, as overexpression of MDM2 inhibits the activity of the tumor repressor p53. MDM2 binders that can block the MDM2:p53 interaction are of key interest for the development of anti-cancer therapies. To test our approach with this PPI target, we synthesized 192 structurally diverse cyclic peptide scaffolds, all based on three random amino acids, one of which contained an amino group for lateral diversification (Supplementary Fig. 9; Scaffolds 2a-f). To increase the chances of identifying binders, we tailored the library to the MDM2 target by including either tryptophan or phenylalanine in all scaffolds, which have been reported to form key interactions in stapled peptides that bind MDM2 and inhibit the MDM2:p53 interaction[33]. We synthesized the cyclic peptide

scaffolds in 96-well plates as described for Library 1, at an average concentration of 12.9 mM and a purity of around 90%. As before, we modified the scaffolds by acylation in a combinatorial fashion at a picomolar scale, this time using 104 carboxylic acids (Supplementary Fig. 10). Thus, we synthesized 100-fold larger library of 19,968 macrocyclic compounds from 192 macrocyclic scaffolds.

We screened Library 2 in 384-microwell plates by dispensing the target protein MDM2 and reporter peptide into the wells containing the macrocycle reactions. The fluorescent reporter peptide binds to the p53-binding interface of MDM2 ($K_d = 0.5\,\mu M$) and its displacement by macrocycles can be measured by a change in fluorescence polarization (FP). We constructed a heat map from the screening results to visualize the extent of reporter peptide displacement from MDM2 (Fig. 4a). We observed a high noise in the FP competition assay, as seen in the heat map as a speckled pattern, which included several isolated dark green spots that we suspected could be false positive hits. However, a group of hits on a horizontal line with higher MDM2 binding stood out, that shared the same scaffold, cyclo[Trp-D3-B4], and we noted the best hits as M6 (acid **14**), M7 (acid **25**), and M8 (acid **91**) (Fig. 4a). We repeated the reactions, which confirmed that the same scaffold and acid combinations yielded the most active products (Supplementary Fig. 11a). We further demonstrated for the hits M6, M7, and M8 that the anticipated macrocyclic structures were the active species in the crude reactions (Supplementary Fig. 11b). A strong vertical line in the heat map in position of acid **98** indicated that this group alone bound MDM2 or interfered with the assay.

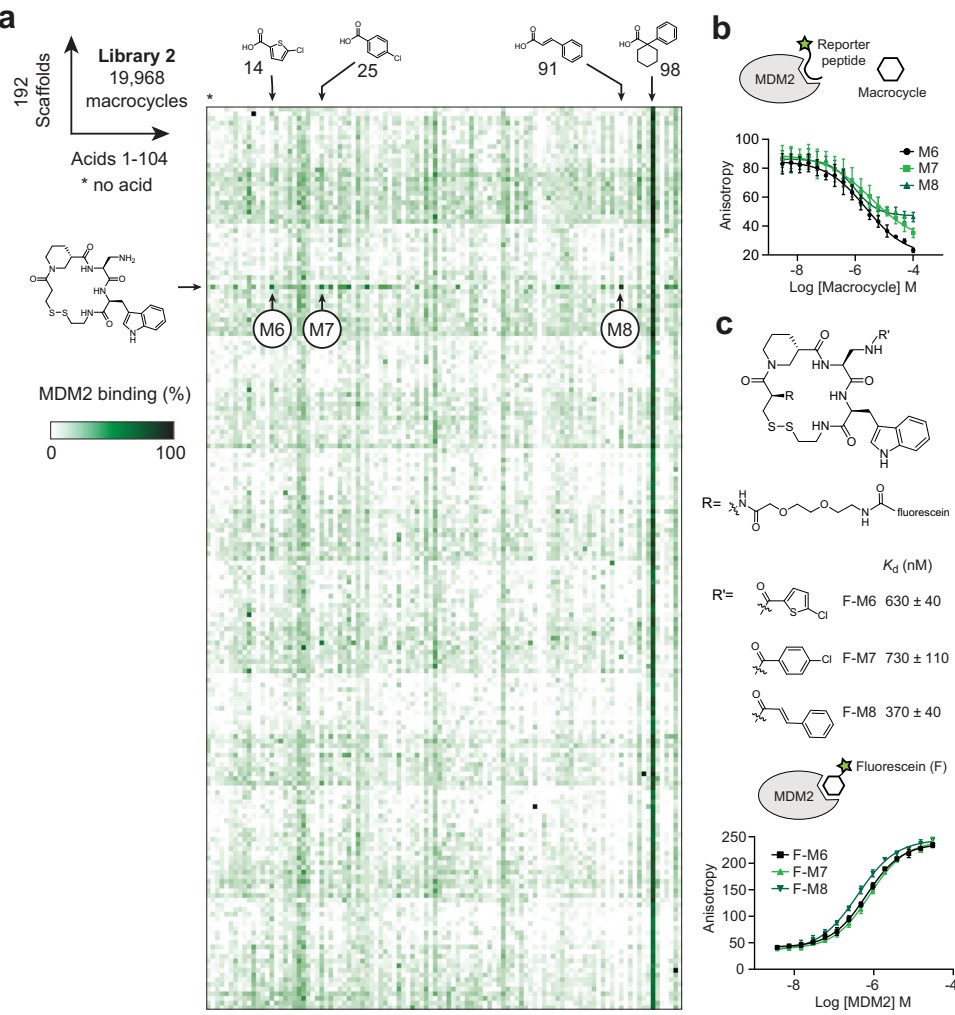

**Fig. 4 Screen against MDM2 and hit characterization. a** The 192 macrocycle scaffolds (Supplementary Fig. 9) were combinatorially acylated with 104 carboxylic acids. The products were screened for displacement of a p53-based fluorescent peptide from human MDM2. Final macrocycle concentrations were 10 μM. For each macrocycle, the MDM2 binding was measured once. The amino acid sequences of the scaffolds are provided in Source Data file. **b** Displacement of the fluorescent reporter peptide from MDM2 by HPLC-purified hit macrocycles M6–M8 measured by FP. Mean values and SDs from three independent displacement assays are shown. **c** Chemical structures of fluorescein-labeled M6–M8 that were measured for affinity to MDM2 in a direct binding assay by FP. Mean values and SDs from three independent measurements of the binding by FP are shown.

The purified macrocycles M6, M7, and M8 efficiently displaced the fluorescent peptide probe from MDM2 (Fig. 4b and Supplementary Fig. 12). As the applied competition assay is not suitable for measuring binding constants below 1 μM due to the requirement of using MDM2 at a minimal concentration of 1.2 μM, we synthesized the three macrocycles as conjugates with fluorescein, and measured the binding affinities in a direct FP assay (Fig. 4c). The $K_d$ values for the conjugates were 630 ± 40 nM (F-M6), 730 ± 110 nM (F-M7), and 380 ± 40 nM (F-M8). The cinnamic acid of F-M8 is an electrophile that could potentially react covalently with nucleophiles such as cysteine side chains in MDM2. Competition binding experiments showed that F-M8 could be displaced and suggested that the macrocycle binds MDM2 only through non-covalent contacts.

**Iterative picomole-scale synthesis and screening**. The facile synthesis of macrocyclic compounds allows for the iterative synthesis of sub-libraries based on hit compounds that can improve binding affinity. To enhance the potency of macrocycle M8 that we identified in the initial screen, we synthesized

63 scaffolds (Fig. 5a) using similar amino acids to those in M8, which were three analogs of nipecotic acid (Nip), three analogs of diamino propionic acid (Dap), and seven analogs of tryptophan (Trp) (3 × 3 × 7 = 63). We then diversified the 63 scaffolds with 14 carboxylic acids, which included repeats from the initial library hits (carboxylic acids **14**, **25**, **91**), as well as related structures that were analogs of cinnamic acid **91**, the carboxylic acid of the macrocycle hit M8 (**105–115**; Supplementary Fig. 10). To identify binders with nanomolar affinity, we screened the 882 macrocycles (63 × 14) at a 13-fold lower concentration (750 nM) than in the first screen, which corresponded to a 30 pmol scale (Fig. 5b). While the most active macrocycles were based on the original scaffold, we identified carboxylic acids in this screen that yielded more potent macrocycles, namely **109** that displaced the reporter peptide by 68% at 750 nM (macrocycle M9), compared to 21% for M8 (acid **91**; Fig. 5b). Although we did not improve upon the scaffold, we gleaned meaningful structure-activity relationship data for the macrocyclic ring from the screen with 63 scaffold and showed that all building blocks were essential.

We subsequently performed a third cycle of library synthesis and screening, where we acylated the nine strongest binding

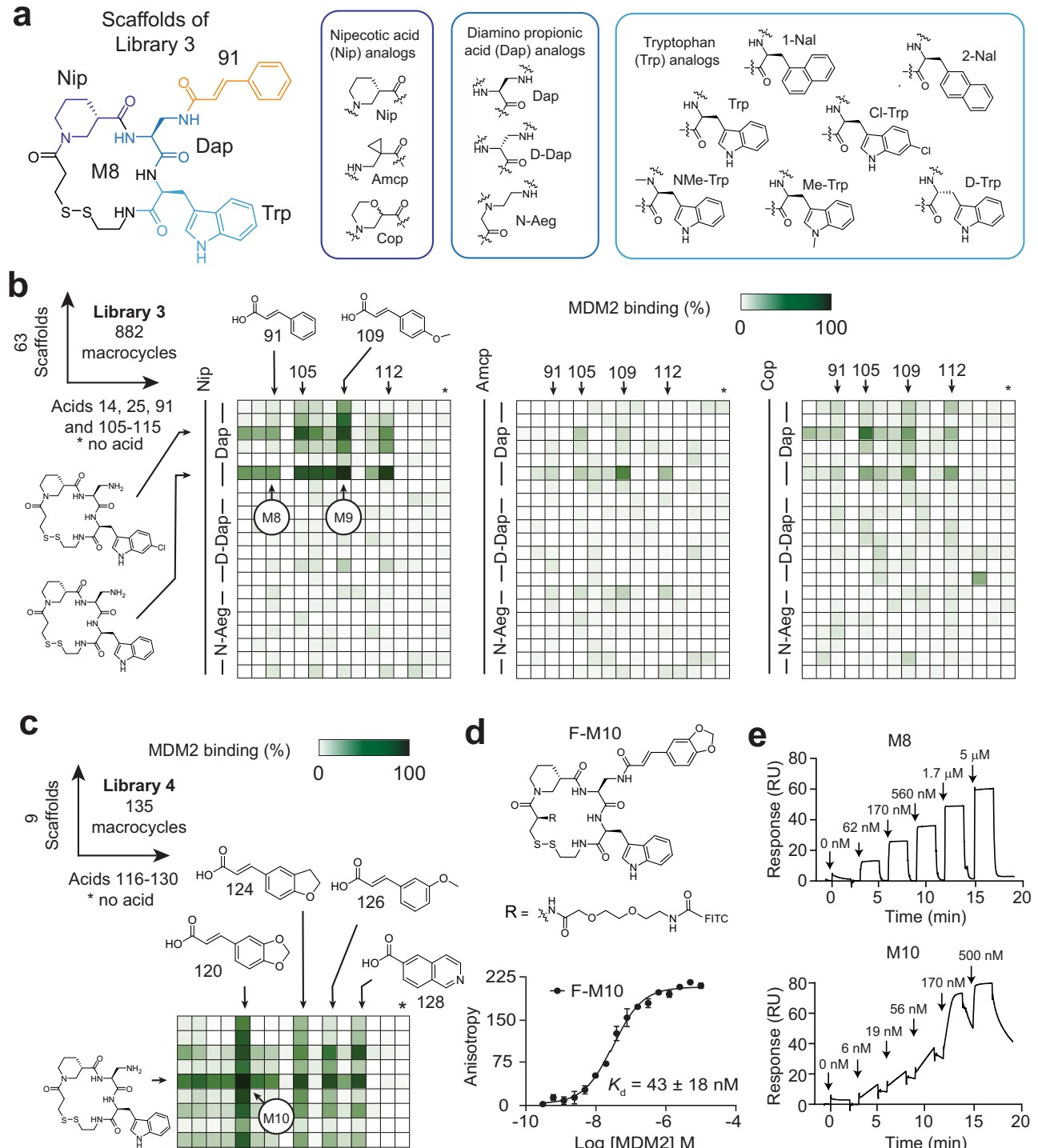

**Fig. 5 Affinity optimization of an MDM2:p53 inhibitor. a** Scaffolds of Library 3 are based on M8 wherein the amino acids shown in blue colors are diversified. Amino acid building blocks are shown in the three frames. **b** Heatmaps of MDM2 binding to the 63 scaffolds that were combinatorially acylated with 15 carboxylic acids at a 30 pmol scale. Binding to MDM2 was measured by displacement assay of the fluorescent peptide probe by macrocycles at a concentration of 750 nM. **c** Screening of Library 4 based on the best nine scaffolds from the previous screens and 15 additional carboxylic acids. **d** Binding of fluorescein-labeled and HPLC-purified macrocycle M10 (F-M10) to MDM2 as measured by FP. Mean values and SDs of three independent measurements of the binding by FP are shown. **e** Binding of unlabeled macrocycles M8 and M10 to MDM2 as measured by SPR.

scaffolds of the previous screen with 15 additional carboxylic acids, which were mostly cinnamic acid derivatives with larger substituents. We subsequently identified M10, which is a macrocycle based on the original scaffold and acylated with acid **120** that displaced the fluorescent probe used, F-M8, to 84% from MDM2 at 750 nM, and more efficiently than the parent

compounds, M8 and M9 (Fig. 5c). We conjugated the macrocycle M10 to fluorescein and measured its binding to MDM2 using FP for a $K_d$ of $43 \pm 18$ nM (Fig. 5d). With the fluorescein conjugate, we also performed a competition experiment with nutlin-3a that binds to a defined hydrophobic pocket of MDM2[34] and is a precursor of nutlin-based clinical candidates. The two ligands did

not compete, indicating that the macrocycle binds to a different site and potentially has a previously undescribed inhibition mechanism (Supplementary Fig. 13). Repetition of the binding measurements of the macrocycles by surface plasmon resonance (SPR) as an orthogonal method showed $K_d$s of $600 \pm 300$ nM (M6), $550 \pm 190$ nM (M7), $169 \pm 93$ nM (M8), and $29 \pm 14$ nM (M10), which confirmed the affinity range found with the FP assay (Fig. 5e and Supplementary Fig. 14).

## Discussion

A major challenge in current drug development efforts is the generation of macrocyclic ligands to challenging protein targets such as PPIs. To address this, we herein detail an approach to rapidly synthesize de novo and screen target-tailored libraries comprising tens of thousands of macrocyclic compounds. We achieved this by: (i) combinatorially diversifying macrocyclic scaffolds by chemical modification of peripheral groups with fragments in microwell plates, (ii) reducing reactions to a pico-mole scale by conducting syntheses with contactless acoustic liquid transfer in nanoliter volumes, and (iii) screening the macrocyclic library products in their crude forms by direct addition of target and screening reagents. Whereas previous methods for synthesizing macrocyclic libraries in microwell plates were limited by the challenging nature of cyclization reactions, we have developed methods for efficient and late-stage diversification chemistry through acylation of peripheral amino groups on macrocyclic parent scaffolds, which increased product yields and purity. By modification of the peripheral amino group, we were able to tap into large numbers of commercially available car-boxylic acids for library diversification. An important consequence of the applied chemistry was our ability to significantly downsize the reaction scale while simultaneously increasing reagent concentrations and accelerating reagent addition by acoustic liquid transfer. The small reagent volumes of 20 nL transferred herein accelerated much the speed of liquid transfer: on the latest generation acoustic dispenser (ECHO 650) used in this work, liquids are transferred in pulses of 2.5 nL, and a transfer of 20 nL thus required only 8 pulses. Assembling 384 reactions in one plate required <5 min, and a library of around 20,000 macrocycles could be produced in only a few hours.

We demonstrated the efficiency of the macrocyclic library synthesis methodology by screening and identifying a high-affinity inhibitor against thrombin ($K_i = 44 \pm 1$ nM). A unique feature of the inhibitor is that it does not contain a charged group, and thus could serve as lead for developing a drug with a higher oral availability than dabigatran etexilate, an approved thrombin inhibitor that is a prodrug. The active form of dabigatran etexilate contains two charges that are shielded by two groups that need to be removed after oral delivery, and this prodrug nature leads to a relatively low oral availability of <6.5%[35]. We also developed a nanomolar inhibitor against the MDM2:p53 interaction ($K_d = 43 \pm 18$ nM), which is an important anti-cancer therapeutic target. The affinity of the macrocycle is rather good considering that it is derived from high-throughput screens and has not yet undergone cycles of improvement by classical medicinal chemistry methods. We should note, however, that the MDM2 binder is still substantially weaker than MDM2 inhibitors that entered clinical studies, as for example idasanutlin (RG7388) (HTRF $IC_{50} = 6$ nM) or navtemadlin (AMG-232/KRT-232) (SPR $K_d = 45$ pM). Compared to a recent attempt to generate macrocyclic MDM2 inhibitors, in which we generated and screened macrocycle libraries by combinatorial cyclization reactions[26], the approach allowed generating a 7-fold larger library and yielded 30-fold stronger MDM2 binders, under-scoring the superiority of the approach.

For the binders of both targets, we found that the carboxylic acids had substantially increased the binding affinity of the cyclic peptide scaffolds, supporting the strategy of developing ligands that bind via both, a macrocyclic scaffold and a peripheral sub-stituent group. In fact, many natural product macrocycles have peripheral groups that contribute to binding, as systematically analyzed and described by Villar et al.[4]. Comparing the herein identified nanomolar hits to macrocycles developed with other high-throughput synthesis and screening methods, we note that they have superior affinities, with for example most macrocycles reported from DNA-encoded libraries having micromolar bind-ing constants[36]. The compounds of the macrocycle libraries synthesized and screened in this work all contain a disulfide bridge, which can get reduced inside cells, leading to linearization and likely degradation and/or inactivation. While we have shown that the disulfide bridge of identified ligands can be replaced by a non-reducible thioether bond, it would be an advantage if "dis-ulfide-free" scaffolds could be used for the library synthesis and screening. Towards this end, we are currently working on a method that offers access to large libraries of thioether-cyclized macrocycles, that is compatible with the herein described late-stage nanomole-scale diversification.

Taken together, the approach opens the door for the gen-eration of massive, target-tailored, and high-purity macrocyclic libraries that are suitable for immediate high-throughput screening. From ongoing work in our laboratory, we envision that even larger libraries of 100,000 compounds can be readily produced using the same methods by expansion to larger multi-well platforms (e.g.1536-well plates) with the goal of exceeding a million elements in the future. Such large libraries may be required for developing binders to particularly challenging targets. The technique of late-stage diversification may also be applied to the generation and screening of other library formats, such as small molecules, DNA aptamers, and even chemically diversified proteins. Important next steps will be the application of the technology to address pressing targets in the pharmaceutical industry, such as generating leads against challenging proteins or PPIs, and to demonstrate in vivo therapeutic uses of macrocyclic ligands.

## Methods

**General considerations.** Unless otherwise noted, all reagents were purchased from commercial sources and used with no further purification. Solvents were not anhydrous, nor were they dried prior to use. The following abbreviations are used: DIPEA (*N,N*-diisopropylethylamine), DABCO (1,4-diazabicyclo[2.2.2]octane), NMM (4-methylmorpholine), HBTU (*N,N,N′,N′*-tetramethyl-*O*-(1H-benzotriazol-1-yl)-uronium-hexafluorophosphate), HATU (*N,N,N′,N′*-tetramethyl-*O*-(7-aza-benzotriazol-1-yl)uronium-hexafluorphosphate), HEPES (4-(2-hydroxyethyl) piperazine-1-ethanesulfonic acid)

**Synthesis of model scaffold 1.** The cyclic peptide model scaffold 1 was synthe-sized using the cyclative disulfide release strategy (CDR) described in the paper Habeshian, S. et al. 2022[31]. The linear peptide precursor was synthesized on a 25 µmol scale in a 5 mL polypropylene synthesis column (MultiSyntech GmbH, V051PE076) using Rapp Polymere HA40004.0 Polystyrene A SH resin (200-400 mesh), 0.95 mmol/gram loading resin and following the procedure described in Habeshian, S. et al. 2022[31]. The peptide was released as follows. For deprotection of the side chains, the resin was incubated with 2 mL of 38:1:1 TFA/TIS/ddH$_2$O v/v/v for 1 h and then washed with $5 \times 4$ mL of DCM. For cyclative peptide release, the resin was treated with 1 mL of DMSO containing 150 mM DIPEA (6 equiv.) overnight. The resin was removed by filtration. The crude mixture was purified by RP-HPLC using a Waters HPLC system (2489 UV detector, 2535 pump, Fraction Collector III), a 19 mm × 250 mm Waters XTerra MS C18 OBD Prep Column C18 column (125 Å pore, 10 µm particle), solvent systems A (H$_2$O, 0.1% v/v TFA) and B (MeCN, 0.1% v/v TFA), and a gradient of 0–25% solvent B over 30 min. The fraction containing the model scaffold was lyophilized and dissolved in DMSO to reach a concentration of 40 mM.

**Acylation of model scaffold 1 by pipetting of reagents.** The model scaffold was acylated at a 40 nmol scale in volumes of 4 µL as follows. The scaffold (20;µL of a

40 mM stock in DMSO) was supplemented with base (20 μL of 160 mM DIPEA dissolved in DMSO), and 2 μL of the mixture were transferred to wells of a PCR plate. The carboxylic acids were prepared as 160 mM stocks in DMSO containing 160 mM DIPEA. Equal volumes of HBTU (160 mM in DMSO) were added to each acid stock, and 2 μL of the resulting active esters (80 mM) were added to the same PCR plate. The reactions were allowed to proceed for 3 h at room temperature. After this time, 1 μL of the reaction was transferred into 99 μL of 100 mM Tris-HCl in water pH 7.5, incubated for 6 h to allow quenching of activated acids with Tris, and the reactions analyzed by LC-MS.

**Acylation of model scaffold 1 by acoustic reagent transfer**. The model scaffold was acylated at an 800 pmol scale in volumes of 80 nL as follows. Scaffold 1 (20 μL of a 40 mM stock in DMSO) was supplemented with base (20 μL of 160 mM DIPEA, 20 μL of 160 mM DABCO, 20 μL of 160 mM HEPES sodium salt or 20 μL of 1 M NMM dissolved in DMSO) and 10 μL of the mixtures were transferred to an ECHO source plate (Labcyte Echo Qualified 384-well Low dead volume micro-plate). The concentrations in the source plate were 20 mM model scaffold and 80 mM DIPEA (4 equiv.), or 80 mM DABCO (4 equiv.), or 800 mM NMM (40 equiv.). The carboxylic acids were prepared as 160 mM stocks in DMSO containing either 160 mM DIPEA, 160 mM DABCO, or 1 M NMM. An equal volume of HBTU (160 mM in DMSO) was added to each acid stock and the active esters (80 mM) were added to the same source plate. The source plate was centrifuged at 950 × g (2000 rpm with a Thermo Heraeus Multifuge 3L-R centrifuge) for 3 min to remove potential bubbles. Using a Labcyte Echo 650 acoustic dispenser, 40 nL of the model scaffold 1 (800 pmol) were transferred to a Nunc 384 well low volume polystyrene plate, followed by 40 nL of the active esters (3.2 nmol, 4 equiv.). The transfers were performed in duplicate in order to have enough material for LC-MS analysis. The plates were sealed, and the reactions were allowed to proceed for 6 h at room temperature. After this time, 8 μL of 100 mM Tris-HCl in water pH 7.5 were added to each one of the duplicate reactions, the duplicates pooled, incubated for 3 h to allow quenching of activated acids with Tris, and the reactions analyzed by LC-MS.

**Acylation of model scaffolds 2–5 by acoustic reagent transfer**. The model scaffolds 2–5 purchased from Enamine were obtained as 1.1 to 1.2 mg powders. The scaffolds were dissolved in 62–88 μL DMSO to obtain 40 mM stocks. The scaffolds were acylated using DABCO as a base, as described for the model scaffold 1 above, with the following differences: 6 h reaction time. Before LC-MS analysis, 720 nL of DMSO was dispensed to each well, then 7.2 μL of 100 mM Tris-HCl in water pH 7.5 was dispensed. Quenching took place overnight.

**Design of scaffolds and amino acid sequences**. The cyclic peptide scaffolds used for Library 1 were prepared by randomly choosing amino acid sequences. The number of different sequences that could theoretically be generated based on the chosen scaffold formats and amino acid building blocks was much larger than the number of scaffolds that were synthesized for Library 1 (384), as described in the following:

Theoretical number of scaffolds for Library 1:

Scaffolds containing diamino acids: 3240
  # scaffold formats (6) × # di-aa (6) × # backbone aa (6) × # side chain aa (15)
Scaffolds containing cysteine: 540
  # scaffold formats (6) × # backbone aa (6) × # side chain aa (15).

For randomly choosing 384 amino acid sequences, all building blocks were assigned an alphanumeric identifier, and every possible permutation was enumerated manually. The peptides were assigned numbers from 1 to 3,780. A random sequence generator (https://www.random.org/sequences/) was then used to re-order the numbers, and the first 384 were chosen for synthesis.

**Preparation of polystyrene-S-S-cysteamine resin for library synthesis**. Polystyrene-S-S-cysteamine resin was prepared in a quantity required for the synthesis of 4 × 96 peptides a 5 μmol scale in 96-well plates, as needed for the synthesis of Library 1 (thrombin screen). A quantity of 589 mg resin with a 0.85 mmol/gram loading (Rapp Polymere HA40004.0 Polystyrene A SH resin, 200-400 mesh), corresponding to a 0.5 mmol scale, was added to each of four 20 mL plastic syringes (CEM, 99.278). In each tube, the resin was washed with 15 mL of DCM and subsequently swelled in 15 mL of 3:7 MeOH/DCM (v/v) for 20 min. The reagent 2-(2-pyridinyldithio)-ethanamine hydrochloride (1.96 grams, 8.8 mmoles, 4.4 equiv.) was dissolved in 21.12 mL of MeOH and 49.28 mL of DCM and 1.53 mL of DIPEA were added. A volume of 17.7 mL of this solution was pulled into each syringe and the syringes were shaken at room temperature for 3 h. The solutions with 2-(2-pyridinyldithio)-ethanamine solutions were the discarded and the resins washed with sequentially with 2 × 20 mL 3:7 MeOH/DCM (v/v) and 2 × 20 mL DMF. The resin of the four syringes was combined into a single syringe as a suspension in DMF, washed with 11.8 mL of 1.2 M DIPEA solution in DMF for 5 min to ensure that all amines were neutral. This solution was then discarded, the resin washed sequentially with 2 × 20 mL DMF, 4 × 20 mL DCM, and then kept under vacuum overnight to yield a free-flowing powder. For the synthesis of peptides for the Library 2 (MDM2 screen), a different batch of commercial resin,

having a loading of 0.95 mmoles/gram, was used (526 mg of resin was added to each of two syringes).

**Peptide library synthesis in 96-well plates**. Automated solid-phase peptide synthesis was performed on an Intavis Multipep RSi synthesizer that has a capacity of four 96-well plates. For the synthesis of peptide for Library 1 (thrombin screen), 565 mg of polystyrene-S-S-cysteamine resin (0.48 mmol cysteamine assuming that thiol groups were quantitatively modified with cysteamine) were added to a 50 mL tube and 20 mL DMF were added. For the MDM2 library, 505 mg of functionalized resin was added instead. The tube was shaken to ensure that the resin was uniformly suspended, and 200 μL (5.88 mg resin, 5 μmoles) were transferred to each well of a 96-well solid phase synthesis plate (Orochem OF 1100). The resin in each well was washed with 6 × 150 μL DMF. Coupling was performed with 53 μL of Fmoc amino acid (500 mM, 5.3 equiv.), 50 μL HATU (500 mM, 5 equiv.), 12.5 μL of N-methylmorpholine (4 M, 10 equiv.), and 5 μL N-methylpyrrolidone. All components were premixed for 1 min, then added to the resin for 1 h reaction without shaking. The final volume of the coupling reaction was 120.5 μL and the final concentrations of reagents were 220 mM amino acid, 208 mM HATU and 415 mM N-methylmorpholine. Coupling was performed twice followed by resin washing with 6 × 225 μL of DMF. Fmoc deprotection was performed twice, each time using 120 μL of 1:5 piperidine/ DMF (v/v) for 5 min. After Fmoc deprotection, the resin was washed with 8 × 225 μL DMF. At the end of the peptide synthesis, the resin was washed with 2 × 200 μL of DCM.

**Removal of protecting groups of library peptides**. The side chains were deprotected with the peptides remaining immobilized on the solid phase, allowing efficient removal of the protecting groups and cleavage reagents. The 96-well synthesis plates were pressed onto a soft 6 mm thick ethylene-vinyl acetate foam pad (Rayher Hobby GmbH, 78 263 01) in order to close the outlets of the wells. The resin in each well was incubated with around 500 μL of 38:1:1 TFA/TIS/ddH₂O (v/v/v) for 1 h. The plates were covered with a polypropylene adhesive seal and weight (1 kg) was placed on top in order prevent detachment of the foam pad and leakage of the deprotection solution. After incubating for 1.5 h, the synthesis plates were placed onto 2 mL deep-well plates (Thermo Scientific, 278752) and the TFA mixture was allowed to drain. The resin in each well was washed with 3 × 500 μL of DCM (added with syringe) and subsequently allowed to air dry for 3 h.

**Cyclative release of library peptides in 96-well plates**. The outlets of the 96-well synthesis plates were closed as described above and 200 μL of 150 mM DABCO in DMSO (6 equiv.) were added to each well. The plates were sealed with an adhesive foil and a weight (1 kg) put on top as described above. After over-night incubation at room temperature, the 96-well synthesis plates were placed onto 2 mL deep-well plates (Thermo Scientific, 278752) and spun at around 200 × g (1000 rpm with a Thermo Heraeus Multifuge 3L-R centrifuge) for 1 min to collect the released disulfide-cyclized peptides.

**Library peptide quantification by absorption**. Absorbance measurements were performed with a Nanodrop 8000 spectrophotometer (Thermo Scientific) at a wavelength of 280 nm using a 10 mm path length. Cleaved peptides containing Trp and D-Trp were diluted 250 fold into water for the thrombin library, and 125 fold into water for the MDM2 library. The Beer-Lambert law was used to calculate the concentration of the peptides. Extinctions coefficient Trp $\varepsilon_{280} = 5,500$ $M^{-1}cm^{-1}$ was used.

**LC-MS analysis**. Peptides were analyzed by LC-MS analysis with a UHPLC and single quadrupole MS system (Shimadzu LCMS-2020) using a C18 reversed phase column (Phenomenex Kinetex 2.1 mm × 50 mm C18 column, 100 Å pore, 2.6 μm particle) and a linear gradient of solvent B (acetonitrile, 0.05% formic acid) over solvent A (H₂O, 0.05% formic acid) at a flow rate of 1 mL/min. Mass analysis was performed in positive ion mode. Data was collected and analyzed using the software of the LC-MS the Shimadzu 2020 instrument.

For the LC-MS analysis, the samples of the various experiments were prepared as follows. For analyzing the acylation proof-of-concept reactions, 160 nL of reaction mixtures were diluted into 16 μL of Tris-HCl buffer pH 7.5 to give a peptide concentration of 100 μM. For analyzing the scaffolds synthesized for Library 1, 1 μL of the DMSO/DABCO eluates were diluted into 80 μL of water to give a cyclic peptide concentration of around 120 μM. For analyzing the scaffolds synthesized for Library 2, 1 μL of the DMSO/DABCO eluates were diluted into 128 μL of water to give cyclic peptide concentration of around 120 μM. For all analyses, 5 μL of the samples were injected, typically using a 0 to 60% gradient of solvent B over 5 min.

**Calculation of physicochemical properties of macrocycles**. The physicochemical properties molecular weight, calculated water/n-octanol partition coefficient (cLogP), number of hydrogen bond donors (HBDs), number of hydrogen bond acceptors (HBAs), polar surface area (PSA), and number of rotatable bonds (NRotB) were calculated using DataWarrior software (www.openmolecules.org), version 5.2.1. The structures of the scaffolds and the carboxylic acids were drawn in

ChemDraw and saved as SMILES strings in SD files, one for the scaffolds and one for the acids. Both SD files were opened in DataWarrior. The "enumerate combinatorial library" functionality was used to define the desired amide bond forming reaction between the macrocycle scaffolds and the carboxylic acids. The following definitions were made: amide was defined as "excluded group". Nitrogen atom was defined as not being part of an aromatic ring, and having a hydrogen atom count >0. The carbon atom next to the amine was defined as being not aromatic, and not containing pi electrons. The starting material and product atoms were then mapped. Following combinatorial enumeration, the desired properties were calculated from the structures.

**Acylation of scaffolds to generate Library 1.** Cyclic peptide scaffolds, in the solvent used to release the peptides from resin (DMSO containing 150 mM DABCO), were transferred to a Labcyte Echo Qualified 384-well Low dead volume microplate (10 μL per well). The concentrations of the cyclic peptide scaffolds were around 8.1 mM in average. Carboxylic acids were dissolved to 160 mM in DMSO containing 160 mM DABCO. An equal volume of HBTU (160 mM in DMSO) was added to each acid stock. The active esters (80 mM) were added to another low-dead-volume source plate. The source plates were centrifuged at $850 \times g$ (2000 rpm with a Thermo Heraeus Multifuge 3L-R centrifuge) for 3 min to remove potential bubbles. Using a Labcyte Echo 650 acoustic dispenser, 20 nL of the scaffolds (160 pmol) were transferred to 384 well low volume polystyrene plates (Nunc, 264705), followed by 20 nL of the active esters (1.6 nmol, 10 equiv.). The plates were sealed, and the reaction was allowed to proceed for 6 h at room temperature. After this time, 5 μL of Tris buffer (100 mM Tris-Cl, pH 7.5, 150 mM NaCl, 10 mM MgCl₂, 1 mM CaCl₂, 0.1% w/v BSA, 0.01% v/v Triton-X100) was dispensed into each well using a BioTek MultiFlo microplate dispenser. The reactions were quenched overnight at room temperature.

**Thrombin inhibition screen.** Thrombin inhibition by the macrocycles of the Library 1 was assessed by measuring residual activity of thrombin in presence of the cyclic peptides at 11 μM average final concentration. The assays were performed in 384-well plates using Tris buffer at pH 7.4 (100 mM Tris-Cl, 150 mM NaCl, 10 mM MgCl₂, 1 mM CaCl₂, 0.1% w/v BSA, 0.01% v/v Triton-X100, and 0.6% v/v DMSO) using thrombin at a final concentration of 2 nM and the fluorogenic substrate Z-Gly-Gly-Arg-AMC at a final concentration of 50 μM. Thrombin (5 μL, 6 nM) in the Tris-Cl buffer described above was added to each peptide using a BioTek MultiFlo microplate dispenser, and incubated for 10 min at room temperature. The fluorogenic substrate (5 μL, 150 μM) in the same buffer was added using the BioTek MultiFlo microplate dispenser, and the florescence intensity measured with a Tecan Infinite M200 Pro fluorescence plate reader (excitation at 360 nm, emission at 465 nm) at 25 °C for a period of 30 min with a read every 3 min. Data was collected and exported using the Tecan Infinite M200 Pro instrument software. The slope of each activity measurement curve was calculated by Excel (version 2016). For the negative controls (20 wells containing DMSO but no macrocycle), an average slope was calculated. The percent of thrombin inhibition was calculated by dividing the slopes and multiplying the results by 100.

**Acylation of scaffolds to generate Library 2.** Cyclic peptides scaffolds, in the solvent used to release the peptides from resin (DMSO containing 150 mM DABCO), were transferred to a Labcyte Echo Qualified 384-well polypropylene microplate (40 μL per well). The concentrations of the cyclic peptide scaffolds were around 12.9 mM in average. Carboxylic acids were dissolved to 184 mM in a 184 mM DMSO solution of DABCO. An equal volume of HBTU (184 mM in DMSO) was added to each acid stock. The active esters (92 mM) were added to the same polypropylene source plate. The source plates were centrifuged at $950 \times g$ (2000 rpm with a Thermo Heraeus Multifuge 3L-R centrifuge) for 3 min to remove potential bubbles. Using a Labcyte Echo 650 acoustic dispenser, 12.5 nL of macrocycles (161 pmol) were transferred to 384 well low volume polystyrene plates (Nunc, 264705), followed by 17.5 nL of the active esters (1.61 nmol, 10 equiv.). The plates were sealed, and the reaction was allowed to proceed for 6 h at room temperature. After this time, 5 μL of Tris buffer was dispensed into each well using a Gyger Certus Flex liquid dispenser, and the reactions were quenched overnight at room temperature.

**MDM2 binding screen.** MDM2 binding by cyclic peptides was assessed by measuring displacement of a fluorescent p53 peptide probe in presence of the cyclic peptides at 11 μM average final concentration. The assays were performed in 384-well plates using PBS buffer at pH 7.4 (100 mM Na₂HPO₄, 18 mM KH₂PO₄, 137 mM NaCl, 2.7 mM KCl, 0.01% v/v Tween 20, and 3% v/v DMSO), MDM2 at a final concentration of 1.2 μM, and the fluorescent p53 peptide probe (FP53, sequence = 5(6)-FAM-GSGSSQETFSDLWKLLPEN) at a final concentration of 25 nM. Premixed MDM2 and FP53 (10 μL, 1.8 mM MDM2, 37.5 nM FP53) in the PBS buffer described above was added to each peptide using a Gyger Certus Flex liquid bulk dispenser, and incubated for 30 min in the dark at room temperature. One fluorescence anisotropy reading was taken with a Tecan Infinite F200 Pro fluorescence plate reader (excitation at 485 nm, emission at 535 nm) at 25 °C. Data was collected and exported using the Tecan Infinite M200 Pro instrument software.

The data was analyzed by Excel (version 2016). The percentage of probe displacement was calculated using to the following formula,

$$\% \text{ probe displacement} = \frac{N - X}{N - P} \times 100 \qquad (1)$$

where $N$ is the average anisotropy of the negative controls (no inhibition), $X$ is the value obtained for each well, and $P$ is the average anisotropy of only the probe.

**Identification of active species in reactions from hits.** The macrocycles identified as hits in the thrombin screen were resynthesized at a 40 nmol scale by reacting 5 μL of 8 mM cyclic peptide scaffolds in DMSO containing 150 mM DABCO with 5 μL of 80 mM carboxylic acid, 80 mM HBTU and 80 mM DABCO for 5 h at room temperature. Remaining activated ester was quenched by addition of 1.25 mL of Tris buffer (100 mM Tris-Cl, 150 mM NaCl, 10 mM MgCl₂, 1 mM CaCl₂) and incubation overnight. The next day, 240 μL of MeCN and 1 mL of water were added, and the reactions were run over a C18 column (7.8 mm × 300 mm Waters NovaPak C-18 column, 60 Å pore, 6 μm particle) on a Thermo Dionex HPLC using solvent A (H₂O, 0.1% v/v TFA) and a 10–80% gradient of solvent B (MeCN, 0.1% v/v TFA) over 20 min, and fractions were collected every minute. Fractions were lyophilized, dissolved in 120 μL of 2% DMSO in water. The activities of products in the fractions were measured using the same assays as described above, but in 96-well plates. 50 μL of each fraction was transferred to a 96-well assay plate (Greiner, 655101), followed by 50 μL of thrombin (6 nM in buffer). After 10 min of incubation, 50 μL of fluorogenic thrombin substrate (Z-Gly-Gly-Arg-AMC, 150 μM in buffer, 1% DMSO) was added and the plates were read and the data processed as described above. Compounds in active fractions were identified by mass spectrometry.

For hits from the MDM2 screen, reactions were performed in the same way but at a 50 nmol scale, and purified with the same method but a 10–80% gradient of solvent B and over 22 min. Fractions were lyophilized and dissolved in 40 μL of DMSO, 160 μL of water was added, and 5 μL of each fraction was transferred to a 384-well plate (Nunc, 264705), and 15 μL of premixed MDM2/FP53 peptide were added (final concentrations: 1.2 μM MDM2, 50 nM FP53, 5% DMSO).

**Crystallization of thrombin with M1.** Human α-thrombin (Hematologic Technologies, Catalog number: HCT-0020) was run over a PD-10 desalting column (GE Healthcare) using buffer containing 20 mM Tris-HCl, 200 mM NaCl, pH 8.0. The human α-thrombin was incubated with the macrocycle M1 at a molar ratio of 1:3 and concentrated to 7.5 mg/mL (protein conc.) using a 3,000 MWCO Vivaspin ultrafiltration device (Sartorius-Stedim Biotech GmbH). Further M1 macrocycle was added during the concentration to ensure a 3-fold molar excess. Crystallization screens with the thrombin:M1 complex were performed at 293 K in a 96-well 2-drop MRC plate (Hampton Research) using the sitting-drop vapor-diffusion method and the Morpheus and LMB crystallization screens (Molecular Dimensions Ltd, Suffolk, UK). Droplets of 600 nL volume (with a 1:1 protein:precipitant ratio) were prepared using an Oryx 8 crystallization robot (Douglas Instruments) and equilibrated against 80 μL reservoir solution. The best crystals were obtained when applying micro-seeding to fresh drops that had been allowed to equilibrate for 2–3 days using the following mixture as precipitant agent: 20 mM sodium formate, 20 mM ammonium acetate, 20 mM sodium citrate tribasic dihydrate, 20 mM potassium sodium tartrate tetrahydrate, 20 mM sodium oxalate, 100 mM MOPS/sodium HEPES pH 7.5, 12.5% (w/v) PEG 1000, 12.5% (w/v) PEG 3350, 12.5% (v/v) MPD.

**Crystallization, data collection and structure determination.** Crystals were mounted on LithoLoops (Molecular Dimensions) and flash-cooled in liquid nitrogen. X-ray diffraction data of human α-thrombin:M1 complex were collected at the i04 beamline of Diamond Light Source Ltd (DLS, Oxfordshire, UK). The best crystals diffracted to 2.27 Å maximum resolution. Crystals belong to the $P2_1$ space group, having unit cell dimensions of $a = 56.25$ Å, $b = 100.57$ Å, $c = 108.90$ Å and $\alpha = 90°$, $\beta = 90.11°$, $\gamma = 90°$. The asymmetric unit contained four molecules. The Matthews coefficient was 2.78 Å³/Da and the solvent content was 48% of the crystal volume. Frames were indexed and integrated using the software XIA2, merged and scaled with AIMLESS (CCP4i2 crystallographic package)[37]. The structure was solved by molecular replacement using the software PHASER[38] and the model 6GWE[19] as template. Refinement was performed using the softwares REFMAC5[39] and PHENIX[40]. From the first cycles of refinement on, a wide electron density corresponding to the bound ligand was clearly visible in the electron density map. Building of the macrocycle was performed using the software Molview and optimized using Phenix eLBOW.[40] The macrocycle was fitted manually using the graphic software COOT.[41] The final model contains 9221 protein atoms, 160 macrocycle atoms, 4 Na⁺ atoms, and 399 water molecules. The final crystallographic R factor reached 0.189 (R_free 0.243). The solvent excluded volume and the corresponding buried surface were calculated using the PISA software and a spherical probe having a 1.4 Å radius. Intra-molecular and inter-molecular hydrogen bond interactions were analyzed by the softwares PROFUNC[42], LIGPLOT + [43], and PYMOL (version 2.3.3). Figures of the structures were prepared using the software PYMOL (version 2.3.3).

**Acylation of scaffolds to generate Libraries 3 and 4**. The cyclic peptide scaffolds required for Libraries 3 and 4 were synthesized as described for those used in Library 2. Due to the presence of many N-methylated amino acids which are more difficult to couple, 200 mM HOAt was applied together with HATU. The scaffolds were diluted to 2 mM in DMSO, and 15 nL were transferred using acoustic dispensing, followed by 15 nL DMSO containing carboxylic acids (40 mM, 20 equiv.), HBTU (40 mM) and DABCO (40 mM). After 6 h reaction at room temperature, 370 nL of DMSO was added to each well, followed by 5 μL of 100 mM Tris-Cl pH 7.4 containing 0.01% v/v Tween20, for quenching overnight. For the MDM2 binding screen, F-M8 was used as a fluorescent probe because of its higher affinity for MDM2, allowing to use the target protein at a lower concentration (720 nM, leading to around 55% bound probe). A volume of 35 μL of PBS buffer at pH 7.4 (100 mM Na$_2$HPO$_4$, 18 mM KH$_2$PO$_4$, 137 mM NaCl, 2.7 mM KCl, 0.01% v/v Tween 20 containing 28.6 nM F-M8 and 823 nM MDM2 were added to each well and the displacement of reporter probe determined as described above. The concentrations of the macrocycles were 750 nM.

**Synthesis of macrocycles at mg scale**. The peptide part of the macrocycles was synthesized on an Intavis Multipep RSi automated solid-phase peptide synthesis at a 25 μmole scale using disposable 5 mL syringes. To each syringe (MultiSyntech GmbH, V051PE076) polystyrene-S-S-cysteamine resin (25 μmole) was added and washed with 6 × 150 μL DMF. Coupling was performed with 210 μL of amino acid (500 mM, 4.2 equiv.), 200 μL HATU (500 mM, 4 equiv.), 50 μL of N-methylmorpholine (4 M, 8 equiv.), and 5 μL N-methylpyrrolidone. All components were premixed for 1 min and then added to the resin for 1 h reaction with shaking. The final volume of the coupling reaction was 465 μL and the final concentrations of reagents were 226 mM amino acid, 215 mM HATU and 430 nM N-methylmorpholine. Each coupling reaction was performed twice. The resin was washed with 2 × 600 μL of DMF. Fmoc deprotection was performed using 450 μL of 1:5 piperidine/ DMF (v/v) for 5 min, and was performed twice. After the Fmoc deprotection, the resin was washed with 7 × 600 μL DMF. At the end of the peptide synthesis, the resin was washed with 2 × 600 μL of DCM.

The side chain protecting groups were removed outside the peptide synthesizer as follows. The resin was incubated with 2 mL of 38:1:1 TFA/TIS/ ddH$_2$O (v/v/v) for 1 h. The deprotection solution was discarded, and the resin washed with 5 × 4 mL DCM. After air drying for 3 h, 1 mL of 150 mM DIPEA in DMSO was pulled into the syringes, and the syringes were shaken overnight at room temperature. The following day, the DMSO solutions were pushed into 50 mL conical tubes.

Carboxylic acids were coupled to disulfide-cyclized peptides by adding 500 μL of premixed acid (100 mM, 2 equiv.), HBTU (100 mM, 2 equiv.) and DABCO (100 mM, 2 equiv.) in DMSO and incubation for 3 h at room temperature. For removing the DMSO before HPLC purification, a volume of 8 mL water was added, the diluted reaction frozen, and lyophilized for 2 days. The contents of the tubes were dissolved in 3 mL of MeCN, followed by addition of 7 mL of water.

The crude mixtures were purified by RP-HPLC using a Waters HPLC system (2489 UV detector, 2535 pump, Fraction Collector III), a 19 mm × 250 mm Waters XTerra MS C18 OBD preparative column (125 Å pore, 10 μm particle), solvent systems A (H$_2$O, 0.1% v/v TFA) and B (MeCN, 0.1% v/v TFA), and typically a gradient of 30–70% solvent B over 30 min.

**Determination K$_i$s of thrombin inhibitors**. Purified thrombin inhibitors (10 mM in DMSO) were diluted to 80 μM in 125 μL of Tris buffer (100 mM Tris-Cl, 150 mM NaCl, 10 mM MgCl$_2$, 1 mM CaCl$_2$) containing 0.1% w/v BSA, 0.01% v/v Triton-X100 and 0.2% DMSO. The macrocycles were diluted two-fold in Tris buffer containing 0.1% w/v BSA, 0.01% v/v Triton-X100 and 1% DMSO in buffer. The thrombin activity was measured in 96-well plates (Greiner, 655101) and the residual activity calculated as described above in the assay used to measure activities of HPLC-separated fractions of screening hits. The residual activity was plotted against the logarithm of the corresponding macrocycle concentrations, and sigmoidal curves were fitted using the following four-parameter equation in GraphPad Prism 6:

$$Y = \text{Bottom} + \frac{\text{Top} - \text{Bottom}}{\left(1 + 10^{(\text{Log}IC50-X) \times \text{HillSlope}}\right)} \quad (2)$$

$K_i$ values were determined from the $IC_{50}$ values using the Cheng-Prusoff equation ($K_m = 168$ μM for thrombin and the applied substrate):

$$K_i = \frac{IC_{50}}{1 + \frac{[S]}{K_m}} \quad (3)$$

**Determination of IC$_{50}$ of MDM2-binding macrocycles**. The concentrations at which MDM2 macrocycles displaced the reporter peptide for 50% of the protein ($IC_{50}$) were determined with the above described fluorescence polarization competition assay. Volumes of 5 μL of purified macrocycles (20 mM in DMSO) were serial diluted two-fold in 100% DMSO in a low dead-volume ECHO source plate. Using acoustic droplet transfer, 150 nL of each dilution was transferred to a 384 well low volume polystyrene plate (Nunc, 264705). A volume of 15 μL of MDM2/ FP53 probe premix (1.2 μM MDM2, 25 nM FP53 probe) in PBS buffer pH 7.4

(100 mM Na$_2$HPO$_4$, 18 mM KH$_2$PO$_4$, 137 mM NaCl, 2.7 mM KCl, 0.01% v/v Tween 20) containing 1% v/v DMSO was added to each well, and incubated for 30 min in the dark. Fluorescence anisotropy was measured as described above. The percentage of bound inhibitor was calculated using the following equation

$$\% \, bound \, inhibitor = \frac{N - X}{N - P} \times 100 \quad (4)$$

where $N$ is the average anisotropy of the DMSO controls, $X$ is the anisotropy value obtained for each well, and $P$ is the average anisotropy of the unbound probe. The $IC_{50}$s were determined by plotting the percent of bound inhibitor against the logarithm of the corresponding macrocycle concentration, and the curves were fitted in GraphPad Prism 6 as described above.

**Synthesis of fluorescein-labeled macrocycles**. Fluorescein-labeled macrocycles were synthesized essentially as described in the mg-scale macrocycle synthesis procedure. For 5(6)-FAM, manual coupling was performed using 4 equiv. of acid (180 mM, 556 μL), 4 equiv. HATU (500 mM, 200 μL), 10 equiv. NMM (4 M, 62.5 μL), all in DMF. The coupling was performed 1 × 2 h, then washed as previously described.

**Determining K$_d$s of fluorescein-labeled MDM2 binders by FP**. Fluorescein-labeled macrocycle stocks (20 mM in DMSO) were diluted to a concentration of 10 μM by adding 0.5 μL into 999.5 μL of PBS. These dilutions were further diluted to a concentration of 100 nM by transferring 10 μL to 990 μL PBS, and 7.5 μL were transferred to wells of a 384 well low volume polystyrene plate (Nunc, 264705). Volumes of 7.5 μL of 2-fold dilutions of MDM2 in PBS were pipetted to the wells. The final concentrations of fluorescent macrocycles were 50 nM. After incubation of the plate for 30 min in the dark at room temperature, the fluorescence anisotropy was measured with a Tecan Infinite F200 Pro fluorescence plate reader (excitation at 485 nm, emission at 535 nm) at 25 °C. Anisotropy was plotted against the logarithm of the corresponding MDM2 concentrations and sigmoidal curves were fitted as described above.

**Synthesis of thrombin inhibitors containing thioether bonds**. Linear peptides containing the three amino acids and the C-terminal cysteamine were synthesized by automated SPPS as described above for the synthesis of mg scale cyclic peptide, but at a 50 μmol scale and using cysteamine 4-methoxytrityl resin (Novabiochem 856087, 200–400 mesh, 1% DVB, 0.92 mmol/gram). To this peptide still on resin, 4-bromobutyric acid (500 μL, 500 mM, 10 equiv.) was coupled manually using N,N'-diisopropylcarbodiimide (DIC, 500 μL, 500 mM, 10 equiv.) as an activating reagent and DMF as solvent. The acid and coupling reagent were premixed for 1 min, then added to the resin (1 h reaction with shaking). The final volume of the coupling reaction was 1 mL and the final concentrations of reagents were 250 mM amino acid, 250 mM DIC. Coupling was performed twice, then the resin was washed with 4 × 4 mL of DMF, then 2 × 4 mL DCM.

Side-chain protecting group removal and cleavage was performed by incubating the resin with 2 mL of 38:1:1 TFA/TIS/ddH$_2$O v/v/v for 1 h with shaking. After this time, 50 mL of cold diethyl ether was added to the solution to precipitate the peptide. The mixture was stored at −20 °C for 30 min, then centrifuged for 30 min at 3800 × g (4000 rpm on a Thermo Heraeus Multifuge 3L-R centrifuge) at 4 °C. The ether was decanted, and the peptide pellet allowed to air dry for 15 min.

The peptide was dissolved in 50 mL of freshly de-gassed 1:4 water/acetonitrile and 200 μL (1.15 mmol, 23 equiv.) of neat DIPEA was added. The cyclization reaction was allowed to proceed at room temperature for 90 mins, then frozen and lyophilized.

Carboxylic acid **14** was coupled as follows. The macrocycle was redissolved in 1 mL of DMSO containing 100 mM DABCO. Carboxylic acids were typically coupled by adding 500 μL of premixed acids (100 mM, 2 equiv.), HBTU (100 mM) and DABCO (100 mM) in DMSO. After 3 h at room temperature, 8 mL of water was added and the tubes were frozen and lyophilized for 2 days to remove DMSO. The contents of the tubes were dissolved in 3 mL of MeCN followed by addition of 7 mL of water. The crude mixtures were purified by RP-HPLC as described above.

**Determining K$_d$s of MDM2 binders by SPR**. Experiments were performed using a GE Healthcare Biacore 8 K instrument. MDM2 (10 μg/mL) was dissolved in 10 mM MES buffer (pH 6.0) and immobilized on three channels of a CM5 series S chip (Cytiva, 29104988) using EDC/NHS amine coupling conditions in running buffer (10 mM PBS pH 7.4, 150 mM NaCl, 3 mM KCl, and 0.005% v/v Tween 20). Typical immobilization level was 6000 to 7000 resonance units (RUs). The reference cell was treated the same way without MDM2. For the measurement of binding kinetics and dissociation constants, five serial dilutions (3-fold) of macrocycles plus a DMSO blank were prepared in running buffer (10 mM PBS pH 7.4, 150 mM NaCl, 3 mM KCl, and 0.005% v/v Tween 20, and 0.5% v/v DMSO) and analyzed in single cycle kinetics mode with contact and dissociation times of 120 s and 60 s, respectively. The data was collected and analyzed using the Biacore 8 K Control Software.

**Reporting summary**. Further information on research design is available in the Nature Research Reporting Summary linked to this article.

## Data availability

Supplementary results (X-ray structure and discussion), three supplementary tables, and 14 supplementary figures are provided in the Supplementary Information. Raw data are provided in a Source Data file. The atomic coordinates of macrocycle M1 bound to thrombin is deposited in the PDB (https://www.rcsb.org) under the accession code 6Z48 (https://www.rcsb.org/structure/6Z48). Source data are provided with this paper.

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

## Acknowledgements

This work was supported by the Swiss National Science Foundation (grants 192368, grant 183443 and NCCR Chemical Biology, all to C.H.). We are grateful to Kaycie Butler and Jing Xin Liang for proofreading the manuscript.

## Author contributions

C.H. and S.H. conceived the strategy and planned the experiments. S.H. performed all the experiments with the following support. M.L.M. and G.K.M. contributed to acylation reaction development, G.S. developed the MDM2 screening assay, M.S. established the ADE infrastructure, Z.B. helped with MDM2 binding studies, C.D.-P. performed the SPR experiments, and J.V., J.B.C. and G.T. helped with ADE liquid transfers. L.C. and A.A. determined the X-ray structure. C.H. and S.H. wrote the manuscript. All authors discussed the results and commented on the manuscript.

## Competing interests

S.H., M.L.M., G.S., G.K.M., M.S., Z.B. and C.H. are inventors of a patent application covering the presented method. C.H. and S.H. are co-founders of Orbis Medicines. The remaining authors declare no competing interests.

**Additional information**

