## [Peer Review File · Nature Communications]

Synthesis and direct assay of large macrocycle diversities by combinatorial late-stage modification at picomole scaleEditorial Note: This manuscript has been previously reviewed at another journal that is not operating a transparent peer review scheme. This document only contains reviewer comments and rebuttal letters for versions considered at *Nature Communications*.

REVIEWERS' COMMENTS

Reviewer #1 (Remarks to the Author):

The authors have addressed my comments and I have no further comments on this manuscript. I have not carefully examined the replies to the other referees, but there appears to be an appropriate feedback and action on those items as well. I recommend proceeding with the acceptance of the publication.

Reviewer #3 (Remarks to the Author):

The authors have addressed all the comments from the three initial reviews satisfactorily. There is just one typo to fix ('electrophile' is misspelled, bottom of page 12 of the revised manuscript). Other than that everything is in order and the reviewers comments have been addressed fully.

Point-by-point discussion of reviewer reports and changes

We would like to thank the reviewers for having read the revised manuscript and for their positive feedback.

Reviewers Comments:

Reviewer #1 (Remarks to the Author):

The authors have addressed my comments and I have no further comments on this manuscript. I have not carefully examined the replies to the other referees, but there appears to be an appropriate feedback and action on those items as well. I recommend proceeding with the acceptance of the publication.

Reviewer #3 (Remarks to the Author):

The authors have addressed all the comments from the three initial reviews satisfactorily. There is just one typo to fix ('electrophile' is misspelled, bottom of page 12 of the revised manuscript). Other than that everything is in order and the reviewers comments have been addressed fully.

Our answer: We have corrected this typo.